# Cardiovascular adaptation to hypoxia and the role of peripheral resistance

**Andrew S Cowburn[1,2]\*, David Macias[1], Charlotte Summers[2], Edwin R Chilvers[2], Randall S Johnson[1,3]\***

[1]Department of Physiology, Development and Neuroscience, University of Cambridge, Cambridge, United Kingdom; [2]Department of Medicine, University of Cambridge, Cambridge, United Kingdom; [3]Department of Cell and Molecular Biology, Karolinska Institute, Stockholm, Sweden

**Abstract** Systemic vascular pressure in vertebrates is regulated by a range of factors: one key element of control is peripheral resistance in tissue capillary beds. Many aspects of the relationship between central control of vascular flow and peripheral resistance are unclear. An important example of this is the relationship between hypoxic response in individual tissues, and the effect that response has on systemic cardiovascular adaptation to oxygen deprivation. We show here how hypoxic response via the HIF transcription factors in one large vascular bed, that underlying the skin, influences cardiovascular response to hypoxia in mice. We show that the response of the skin to hypoxia feeds back on a wide range of cardiovascular parameters, including heart rate, arterial pressures, and body temperature. These data represent the first demonstration of a dynamic role for oxygen sensing in a peripheral tissue directly modifying cardiovascular response to the challenge of hypoxia.

DOI: https://doi.org/10.7554/eLife.28755.001

**\*For correspondence:**
asc32@cam.ac.uk (ASC);
rsj33@cam.ac.uk (RSJ)

**Competing interests:** The authors declare that no competing interests exist.

## Introduction

Vertebrates confront the world chiefly through the skin. As a reflection of how different vertebrates adapt to the environment, skin structure varies to a significant degree from one vertebrate class to another, and even within classes. One of the key differences in skin function found amongst vertebrates is the degree to which it responds to environmental variations; in particular, the extent to which these changes in turn affect systemic organismal physiology. Mammalian skin is amongst the most complex of all the vertebrate forms of skin. It also varies significantly in its form in the different orders of mammalia, with variations chiefly following adaptations to the external environment.

A key physiological challenge faced by most multicellular organisms is variation in oxygen supply. This is a common challenge to animals that obtain their oxygen from surrounding water, particularly those that live in fresh water; it can apply also to mammals at high altitudes, or during movement into a small den or confined space. Hypoxia also occurs in various tissues if there are high levels of oxygen demand, e.g., in skeletal muscle during exercise. Work undertaken over the last 50 years has shown that acute systemic hypoxia in mice, rats, rabbits, chickens, dogs, sheep and humans causes acute tachycardia and hypertension (*Korner and Edwards, 1960*; *Butler, 1967*; *Yasuma and Hayano, 2000*; *Campen et al., 2004*; *Campen et al., 2005*; *Walsh and Marshall, 2006*; *Heinonen et al., 2016*; *Giussani et al., 1993*; *Fletcher, 2000*). Prolonged hypobaric hypoxia in humans has also been shown to cause tachycardia and hypertension (these studies have primarily been undertaken in high altitude environments or models thereof)(*Schultz et al., 2014*; *Calbet, 2003*; *Naeije, 2010*; *Hainsworth et al., 2007*; *Vogel and Harris, 1967*).

The skin has an extensive vasculature, which is known to be responsive to shifts in oxygen availability (*Durand et al., 1969*; *Weil et al., 1969*; *Kuwahira et al., 1993*; *Minson, 2003*). Keratinocytes

**eLife digest** Diseases of the heart and blood vessels are linked with high blood pressure. The causes of most cases of high blood pressure are unknown, but it is often accompanied by the reduced flow of blood through small vessels in the skin and other parts of the body that are a long way from the heart. It is not clear why this change happens or why it tends to get worse over time in people with high blood pressure that has not been treated.

Previous research has shown that when a tissue is starved of oxygen, blood flow to that tissue will increase. The HIF family of proteins help to increase blood flow to tissues in these low-oxygen situations. To investigate what role the skin plays in the flow of blood through small vessels, Cowburn et al. exposed mutant mice that cannot produce certain HIF proteins specifically in the skin to low-oxygen conditions. The experiments show that mice lacking either HIF-1α or HIF-2α in the skin have altered responses to oxygen starvation that affected their heart rate, blood pressure, skin temperature and general levels of activity. Mice lacking specific proteins controlled by the HIFs also responded in a similar way.

Cowburn et al. also demonstrated that the way normal healthy mice respond to oxygen starvation is more complex than previously thought. Blood pressure and heart rate rise in during the first ten minutes. This is followed by a period of up to 36 hours where blood pressure and heart rate decrease below normal levels. By around 48 hours after exposure to low levels of oxygen, blood pressure and heart rate recover, returning to normal levels. Loss of the HIF proteins or other proteins involved in the response to oxygen starvation specifically in the skin affect when this process starts and how long it takes.

These findings suggest that the responses of the skin to environmental challenges may have substantial effects on the how the heart pumps blood around the body. More studies are needed to understand how the HIFs and other proteins may contribute to high blood pressure and diseases affecting the heart and blood vessels.

DOI: https://doi.org/10.7554/eLife.28755.002

in low oxygen release nitric oxide to increase blood flow and thereby improve perfusion (*Cowburn et al., 2013*; *Pucci et al., 2012*). Of interest, it has been shown that in both rodents and humans the basal epidermis is hypoxic relative to the underlying dermal tissue, and displays constitutive stabilisation of hypoxia inducible transcription factor-(HIF)α proteins (*Cowburn et al., 2013*; *Cowburn et al., 2014*; *Peyssonnaux et al., 2008*; *Boutin et al., 2008*).

HIF transcription factors initiate the transcription of multiple genes involved in oxygen homeostasis, including a number that regulate vascularisation and metabolism (*Semenza, 2003*; *Semenza, 2009*; *Pouysségur et al., 2006*; *Formenti et al., 2010*; *Djagaeva and Doronkin, 2010*; *Hubbi et al., 2014*). We have previously shown that a dichotomous regulation of nitric oxide is conferred by HIF-1α and HIF-2α regulation of the NOS2 and arginase genes (*Semenza, 2003*; *Jung et al., 2000*) and have now documented this in a number of tissues and cell types (*Cowburn et al., 2013*; *Takeda et al., 2010*; *Branco-Price et al., 2012*; *Cowburn et al., 2016*). The HIF-1α/NOS2 and HIF-2α/arginase pathways strongly influence tissue L-arginine consumption and NO generation (*Cowburn et al., 2013*; *Takeda et al., 2010*; *Branco-Price et al., 2012*).

We have also shown that keratinocyte selective deletion of HIF-1α reduces epidermal NO levels, whereas loss of HIF-2α suppresses arginase expression/activity, increasing local concentrations of NO (*Cowburn et al., 2013*). Hence, in murine skin, cutaneous NO levels are determined by differential HIFα isoform expression which affect both local vascular resistance and systemic blood pressure. Little is known, however, concerning how a localized tissue response to hypoxia affects the cardiovascular system more generally.

The literature on the response of animals to systemic hypoxia and its cardiovascular effects is limited almost exclusively to studies under anaesthesia. Hence, almost all previous studies have focused on the first ten minutes of the hypoxic response. Radio-telemetry in non-anesthetised non-restrained animals allows a more rigorous evaluation of the changes in hemodynamic variables during the acclimation to environmental hypoxia (*Kawaguchi et al., 2005*). The work described here demonstrates for the first time that systemic hypoxia induces a tri-modal response: with an initial tachycardia and

hypertensive phase that lasts approximately 10 min, followed by an immediate and profound drop in heart rate and blood pressure that can last as long as 24 hr, followed by a recovery to close to a normoxic state. In this context, we show that peripheral resistance and oxygen sensing, in this case in the skin, plays an essential role in this systemic cardiovascular response. These data demonstrate that peripheral tissues can play a fundamental role in cardiovascular adaptation to hypoxia.

## Results

### Loss of HIFα isoforms in the epidermis differentially affects blood pressure and skin temperature across diurnal cycles

In this study we utilise mouse strains with conditional alleles of *Hif1a* (*Ryan et al., 2000*), *Epas1*(HIF-2α)(*Gruber et al., 2007*), the HIF-1α target gene, nitric oxide synthase-2 (*Nos2*) and the HIF-2α target gene, arginase-1(*Arg1*)(*El Kasmi et al., 2008*), crossed to mouse strains expressing Cre recombinase under the control of the keratin 14 (*Krt14*) promoter (*Vasioukhin et al., 1999*). All mice are extensively backcrossed (more than 10 generations after selection by strain-specific SNP analysis) into the C57/Bl6J strain background. This promoter drives the excision of the gene of interest where the K14 promoter is active, that is, within basal layers of the epidermis and in restricted numbers of cells in a small number of other epithelial tissues, chiefly the oesophagus and salivary and mammary glands (*Vasioukhin et al., 1999*).

Continuous monitoring of unrestrained, non-anesthetised mice via radio-telemetric catheterization showed that although resting heart rates were unchanged in all mutants relative to wild type controls (n = 7) through diurnal variation (*Figure 1A*), there was a constitutive hypotensive state in K14cre-HIF-2α mice (n = 7) and NOS2 mutants (n = 6) (*Figure 1B*) and a strong trend towards a hypertensive state in K14cre-HIF-1α mice.

Cutaneous temperatures were also monitored by radiotelemetry (*Figure 1C*). Here, we saw that cutaneous temperatures in K14cre-Arg-1 mutants was significantly higher than littermate controls and trend lower in HIF-1α mutants. These data confirm our previous observations using tail-cuff occlusion and infra-red analysis of skin temperature (*Cowburn et al., 2013*). Analysis of physical activity showed no significant difference in the movement levels between the mutant groups and littermate control mice, suggesting these differences in temperature do not result from variable activity (*Figure 1D*).

### Cardiovascular responses to 48 hr of hypoxia: brief initial hypertension and tachycardia, followed by hypotension and bradycardia

The effects of hypoxia in mammals have been studied in humans and a range of animals(*Korner and Edwards, 1960*; *Butler, 1967*; *Yasuma and Hayano, 2000*; *Heinonen et al., 2016*; *Kawaguchi et al., 2005*). Curiously, published measurements to date indicate that almost all studies examining the cardiovascular effects of hypoxia have been carried out either over long time frames, that is, weeks and months, as part of studies of high altitude adaptations, or over very short time frames of approximately 5 to 10 min, typically under anaesthesia. To better understand how hypoxia at varying levels affects the cardiovascular parameters of blood pressure, heart rate, and cutaneous temperature in mice over time, we subjected wild-type (C57/Bl6) mice with implanted radiotelemetric reporters to 15%, 12%, and 10% normobaric oxygen over 48 hr, followed by 24 hr of recovery at 21% normobaric oxygen. The transition to low oxygen levels occurred in each case at the dark-light interphase (dark cycle equates to the shaded regions in the graphs).

As noted above, the acute cardiovascular response to hypoxia has been described in a number of animal models. As previous investigators have documented, there was an acute response in the first 10–15 min, with increased ventilation (160BrPM to 250BrPM), and a hypoxic pressor reflex, with blood pressures increasing from means of 123/88 mmHg (baseline) to 143/102 mmHg within the first 10 min (*Figure 2A–B*). Following this brief initial response, there was a severe drop in systemic blood pressures at the 12% and 10% oxygen levels, to means of 87 mmHg SBP and 57 mmHg DBP at 120 min for 10% oxygen (*Figure 2C–D*). Heart rates peak at 13 min in the lowest level of hypoxia, from 606 BPM (baseline) to 717 BPM at the lowest levels of oxygen, followed immediately by a decline to 285 BPM (a 53% drop from baseline) at 120 min post-transition (*Figure 2E–F*).

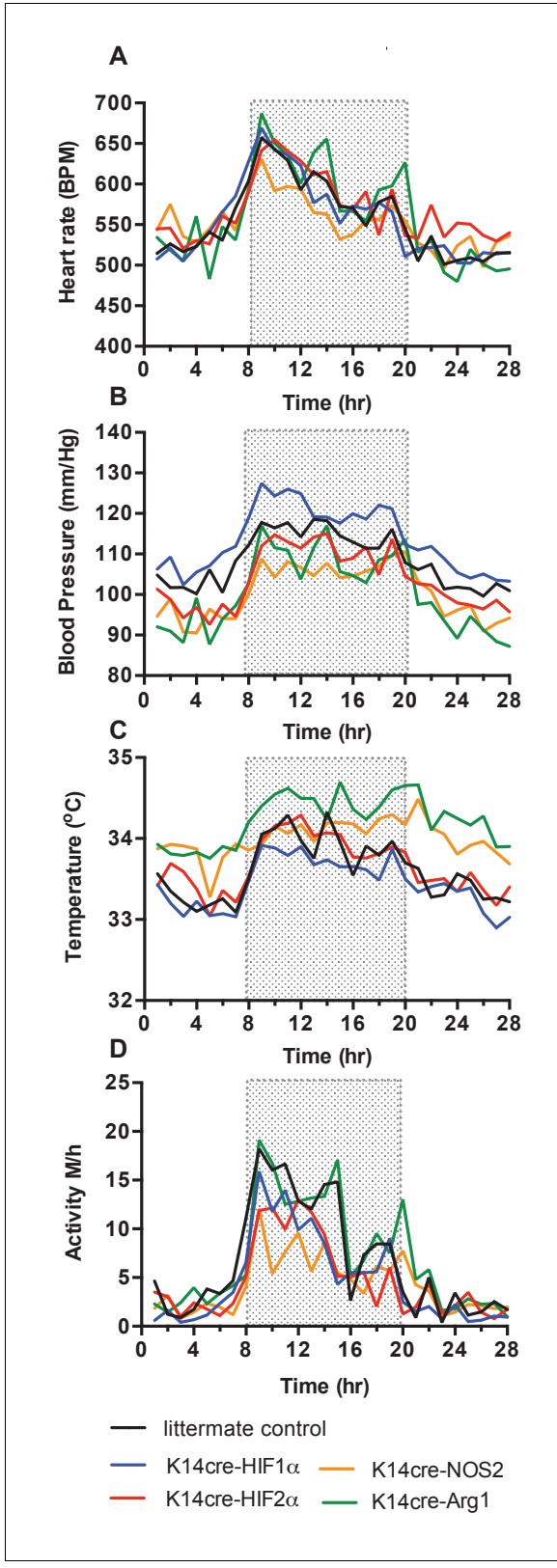

**Figure 1.** Baseline cardiovascular radio-telemetry parameters from littermate and K14cre- mice. Baseline parameters were recorded continuously over 28 hr in a 12hr-light/12hr-dark (shaded area) cycle. Data lines for littermate (black, n = 10), K14cre-HIF-1α (blue, n = 9), K14cre-HIF-2α (red, n = 7), K14cre-Arg1 (red, n = 5), K14cre-NOS2 (orange, n = 6) (**A**) show heart rate as mean beads per minute (**B**) mean blood pressure shown as mm/Hg,
*Figure 1 continued on next page*

*Figure 1 continued*

(C) temperature shown as mean degree's Celsius and (D) physical activity shown as mean meters per hour. Statistical analysis: Area under curved was determined for each mouse per group and each group was further analysed by mann-Whitney test. Mean blood pressure K14cre-HIF-2α (p=0.0274) and K14cre-NOS2 (p=0.0120) and skin temperature K14cre-Arg1 (p=0.0350) compared to littermate control.
DOI: https://doi.org/10.7554/eLife.28755.003

Analysis of subcutaneous temperature has previously been used as an indirect measurement of peripheral vascular resistance(*Schey et al., 2009*). We found that skin temperature rapidly decreased from 33.7°C (baseline) to 28.9°C (−4.8°C) 180mins post hypoxic challenge (*Figure 2G*), suggesting an increase in peripheral vascular resistance. Physical activity also declined from 5.3 m/hr to 0.2 m/hr in the 120 min following the introduction of 10% oxygen. This change, similar to those seen in blood pressure and heart rate, demonstrates that lowered levels of oxygen cause a triphasic cardiovascular response, with the degree and duration of this response dependent on the level of oxygen.

At the lowest $FiO_2$ studied, 10% oxygen, there was an initial phase of response lasting 10–20 min that includes increased blood pressures and heart rates; then a longer depression, resulting in profound hypotension and bradycardia; followed finally by a chronic acclimation phase, with gradually recovering blood pressures, heart rates, temperatures and activity levels. We defined recovery as the point at which the animal reached 90% of heart rates and blood pressures seen at the same time of the diurnal cycle in normoxia; with this definition, recovery occured in 36 hr at 10% oxygen levels in wild type controls. In wild type mice this tri-phasic response is readily apparent at the two lower concentrations of oxygen (*Figure 2*). We chose to use a 10% oxygen challenge for the remainder of our experiments, as this level of hypoxia had the most pronounced (and thus easily dissected) phases of response, and yet still showed a return to normoxic cardiovascular parameters over time.

## Loss of HIF-1α in the epidermis retards cardiovascular adaptation to hypoxia

As shown above, mice lacking HIF-1α, HIF-2α, NOS2 and Arg-1 expression in the epidermis show significant changes in systemic blood pressures and cutaneous temperatures relative to littermate control animals. All of these genes are involved in cellular adaptations to hypoxia. However, it is unclear how this cellular adaptation leads to more general tissue-specific adaptations to hypoxia. Even less well understood is how a peripheral tissue response might affect systemic responses to hypoxia. To undertake an analysis of that question, using the skin as a model for a peripheral tissue with a substantial vascular bed, we analysed the effect of loss of these hypoxia-responsive genes on the cardiovascular hypoxic response. We began with an analysis of the role of HIF-1α.

As described above and shown in *Figure 1*, there is a decrease in skin temperature in mice lacking HIF-1α in keratinocytes. In these mice, the onset of 10% environmental hypoxia causes a sharper drop in skin temperatures (*Figure 3A*) than that seen in littermate control animals, and temperatures of the skin in HIF-1α mutants remain well below those of littermate mice in the second phase of hypoxic response; they do not recover until the restoration of normoxia (for statistical analysis see Figure 9A). Infra-red analysis of skin temperature was also used to monitor peripheral vascular perfusion during hypoxic challenge. The initial drop in skin temperature was comparable between K14cre-HIF-1α mutant and littermate control mice, as shown in *Figure 3B*.

As described above, the HIF-1α mutants are essentially hypertensive under normoxic conditions, and this hypertension is exaggerated in the first phase of hypoxic response (*Figure 3C*, for statistical analysis see Figure 9B). However, in the second phase there is a larger drop in systolic and diastolic pressures relative to littermate control mice (*Figure 3D*, for statistical analysis see Figure 9C). This resolves rapidly, and although the subsequent systolic pressures in mutants are similar to those in littermate control animals, the diastolic pressures in the mutants begin to elevate. This elevated diastolic pressure begins in the first hour of hypoxia, and gradually increases throughout the 48 hr of hypoxic exposure (*Figure 3D*). An elevation in systolic pressures in HIF-1α mutants does not reappear until the animals are restored to normoxia. Elevation of diastolic pressures is indicative of an increase in peripheral resistance, and this is consistent with the lowered skin temperatures described above (*Figure 3A*).

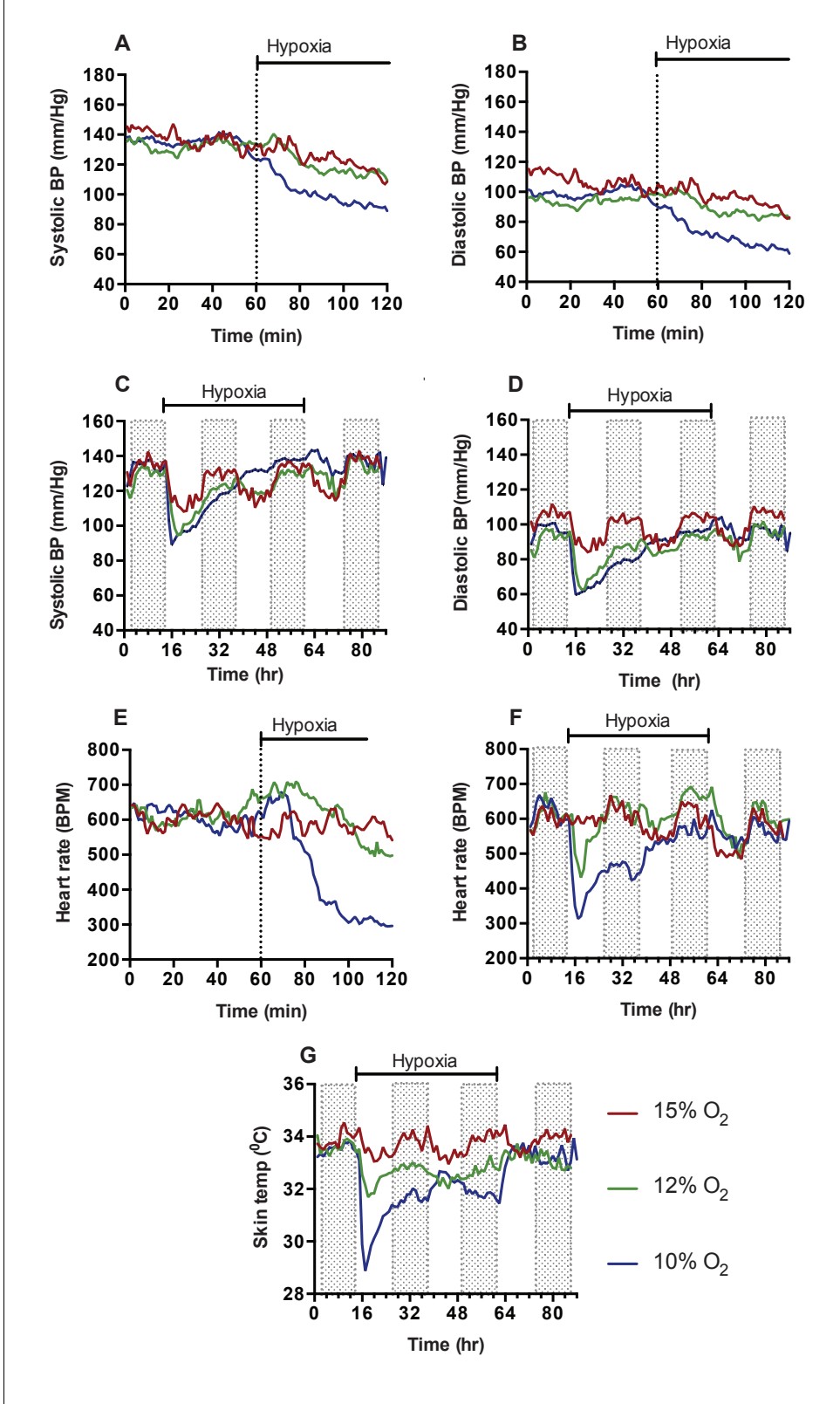

**Figure 2.** Cardiovascular response to graded hypoxia in wild-type (C57/Bl6) animals. Mice were individually housed for 14 hr in normoxia before being exposed to hypoxia (15% $O_2$ (dark red, n = 4), 12% $O_2$ (green, n = 5), 10% $O_2$ (blue, n = 8)) at the light-dark interphase for 48 hr. Mice then recovered for a further 24 hr in normoxia. (A) Acute analysis of systolic and (B) diastolic blood pressure shown as mean (mm/Hg). Data shows 60 min of normoxia followed by 60 min of hypoxia. The dotted line depicts point of transition from normoxia to hypoxia. (C) Chronic systolic and (D) diastolic acclimation to
*Figure 2 continued on next page*

*Figure 2 continued*

hypoxia. Data shown as mean (mm/Hg) (shaded area depicts dark cycle) (E) Acute and (F) chronic heart rate acclimation to hypoxia. Data shown as mean beats per minutes. (G) Chronic Skin temperature acclimation to hypoxia. Data shown as mean degrees Celsius.

DOI: https://doi.org/10.7554/eLife.28755.004

In *Figure 3E and F*, the acute and chronic changes in heart rate in littermate control and mutant animals can be seen (for statistical analysis see Figure 9D and E). The chief difference seen in the HIF-1α mutants is an increase in heart rate that is likely required to sustain systolic pressures; this coincides with the relative increase in diastolic pressures in HIF-1α mutants. Unlike the increased diastolic pressure, this tachycardia resolves following approximately 36 hr of hypoxia, indicating cardiac acclimation has occurred at that point. The chronic change in heart rate in HIF-1α mutants appears to coincide with an increase in whole-body metabolic activity between 12–36 hr hypoxia, as determined by $VO_2$ and $VCO_2$ analysis (*Figure 3G and H*). Normal metabolic activity is only restored when the animals are returned to normoxia.

Physical activity is reduced in littermate control and mutant animals during the initial 12 hr of hypoxia. Activity slowly increases during the acclimation period, but does not fully recover until the animals are restored to normoxia (*Figure 3H*). Overall, loss of HIF-1α in the epidermis appears to delay the cardiovascular changes that occur in response to hypoxia; this is correlated with an increase in peripheral resistance and in heart rates.

## Loss of HIF-2α in the epidermis modifies cardiovascular adaptation to hypoxia

We have proposed a model for hypoxic responses involving HIF isoform activity and nitric oxide homeostasis in the skin that argues that hypoxia results in an initial HIF-1α/NOS2/NO mediated increase in vascular diameter, and resulting increases in perfusion; followed by a HIF-2α/Arginase-induced reduction of intracellular L-arginine (*Cowburn et al., 2013*). As L-arginine is necessary for NO production, this would suppress NO production by NO synthases, and ultimately cause vasoconstriction.

This model for HIFα isoform action in hypoxic vascular tissues fits the observations described above for the loss of HIF-1α in the skin. In animals lacking HIF-2α in the epidermis, adaptation of skin temperatures in the first 12 hr of hypoxia follows the predictions of this model (*Figure 4A*), that is, we see elevated skin temperatures in the HIF-2α mutants relative to littermate control animals. These data are supported by infra-red analysis of skin temperature during short term hypoxia exposure (*Figure 4B*). The recorded drop in skin temperature is significantly less in the HIF-2α mutants relative to littermate control animals. Representative photomicrographs show the rapid drop in skin temperature during hypoxia exposure. However, after 24 hr, skin temperatures in mutants drop relative to littermate control animals, and remain lower than those of littermate control animals for 12 hr, before rising again to levels similar to those of littermate controls. Upon restoration of environmental normoxia, skin temperatures in mutants return to levels slightly higher than those seen in littermate control animals, that is, similar to those seen at steady state (for statistical analysis see Figure 9F).

As shown in *Figure 1*, mean blood pressures of HIF-2α mutants are significantly lower than those of littermate control animals. For the first 24 hr of hypoxia, HIF-2α mutants show elevated systolic and diastolic pressures that are evident in the first minutes of hypoxic exposure (*Figure 4C*, [for statistical analysis see Figure 9G] and *Figure 4D*, [for statistical analysis see Figure 9H]). Here again, there is a shift at 24 hr, and although diastolic pressures at that point are similar to those seen in littermate control animals, systolic pressure drops relative to littermate controls. The systolic blood pressures remain substantially lower until normoxia is restored, when they return to steady state and mildly hypotensive levels. Finally, at the return of normoxia, diastolic pressures rise to levels substantially higher than those seen in littermate control animals for approximately 12 hr, before finally dropping again 24 hr after the restoration of normoxia. This effect on both systolic and diastolic pressures indicates that changes in the skin affect both traditionally understood changes in peripheral resistance, that is, changes that result in altered diastolic pressures, and other changes that can influence systolic pressures.

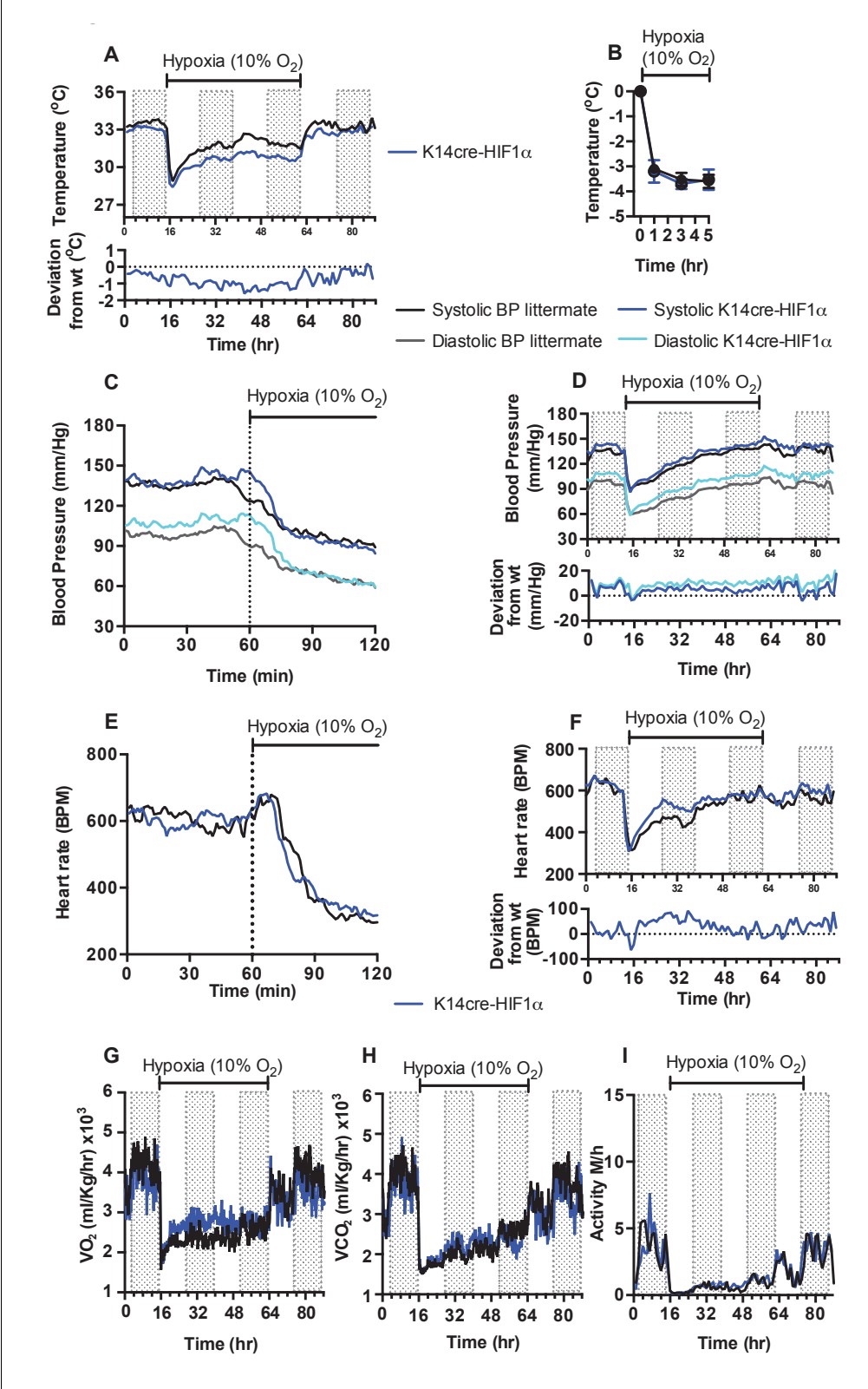

**Figure 3.** Keratinocyte deletion of HIF-1α affects cardiovascular acclimation to hypoxia. Mice were individually housed for 14 hr in normoxia before being exposed to hypoxia (10%) at the light-dark interphase for 48 hr. Mice then recovered for a further 24 hr in normoxia. (A) Chronic skin temperature acclimation in littermate (black line, n = 7) and K14cre-HIF-1α (blue line n = 8). Data shown as mean degree's Celsius (shaded areas depict dark cycle). Lower line graph shows temperature deviation of K14cre-HIF-1α from littermate during the procedure. (B) Skin temperature was monitor by an infrared
*Figure 3 continued on next page*

*Figure 3 continued*

thermal imaging camera before and during hypoxia (10% O$_2$) exposure. Analysis of thermal infrared images suggests that K14cre-HIF-1α (n = 5) does not substantially impact skin temperature when compared to littermate (n = 5) controls during the first 5 hr of hypoxia. (C) Acute and (D) chronic analysis of systolic and diastolic blood pressure in littermate (black-line and grey-line n = 8) and K14cre-HIF-1α (dark blue line and light blue line n = 7) respectively. Data shown as mean (mm/Hg). Lower line graph shows blood pressure deviation for both chronic systolic (dark blue) and diastolic (light blue) BP. (E) Acute and (F) chronic analysis of heart rate in littermate (black line n = 6) and K14cre-HIF-1α (dark blue line n = 8). Data shown as mean beats per minute. Lower line graph shows chronic heart rate deviation of K14cre-HIF-1α from littermate. (G) Analysis of metabolic activity during the chronic acclimation of littermate (black line, n = 7) and K14cre-HIF-1α (dark blue line n = 5) to hypoxia. Data shown as mean oxygen consumption VO$_2$ and (H) carbon dioxide generated VCO$_2$ ml/Kg/hr. (I) Analysis of physical activity in littermate (black line n = 6) and K14cre-HIF-1α (dark blue line n = 7) during acclimation to chronic hypoxia. Data shown as mean activity meters/hour.

DOI: https://doi.org/10.7554/eLife.28755.005

The effect of hypoxia on the heart rate of the epidermal HIF-2α knockout animals is strikingly altered when compared to the effect on littermate control animals. Hypoxia has virtually no effect on heart rate in the HIF-2α mutant animals (*Figure 4E*, for statistical analysis see Figure 9I); they show a diurnal variation of heart rate that is essentially the same as that seen in animals in normoxic conditions (*Figure 4F*, for statistical analysis see Figure 9J). This is in comparison to the approximate 50% decline in heart rates seen in littermate control animals after two hours at 10% oxygen. Whole-body metabolic activity (VO$_2$ and VCO$_2$) in HIF-2α mutant animals is also significantly increased above littermate control rates between 12–24 hr hypoxia (*Figure 4G and H*). This increased metabolic activity does not return to levels seen in littermate control animals until the mice are returned to normoxia. Physical activity, similarly to the patterns described above, is greatly reduced in littermate and mutant animals during the initial 12 hr of hypoxia. However, activity of HIF-2α mutant animals recovers substantially between 12–48 hr, although it does not fully recover until the animals are restored to normoxia (*Figure 4I*).

These results indicate a complex response of the skin to systemic hypoxia, and also demonstrate that that complexity is mirrored in the effect of those responses on the cardiovascular system. The crossover seen in skin temperature and blood pressure at 24 hr also indicates that there is likely a phase of adaptation at that time point revealed by the HIF-2α deletion, and related to a temporally specific action of HIF-2α in the adaptation and response to hypoxia.

## Expression patterns and levels of NO in peripheral tissues during hypoxia indicate tissue-specific variation in response

Given the complex role of HIF-2α in the epidermis, and our hypothesis that this could be determined by the differential effect of HIF-1α and HIF-2α on NO homeostasis, we next analysed the overall levels of NO metabolites induced by hypoxia in specific tissues following hypoxic exposure (*Figure 5*). As can be seen in *Figure 5A*, there was a steep drop in plasma NO metabolites over the first day of hypoxia, followed by a more gradual decline over the following 3 days of exposure (*Figure 5A*). However, as shown in *Figure 5B*, in skin, and in *Figure 5C*, in lungs, differing levels of expression of hypoxia-sensitive genes involved in the NO synthesis pathway are evident. Of note, the skin shows an initial spike in arginase-1 and -2 expression in the first 24 hr of exposure, which then declines. Skin NOS2 expression over the same time is marginally higher, however. In the lung there is no substantial increase in NOS2 or arginase-1 or -2 expression until after two days of exposure to hypoxia.

The levels of NO metabolites seen in the skin and lung reflect the expression levels of arginase and NOS2 mRNA, with an increase seen in the skin at 24 hr and a subsequent decline to baseline levels after 2 days of hypoxia (*Figure 5D*). This is in contrast to the lung, which shows a gradual decline in NO metabolites at 48–72 hr (*Figure 5E*). These data indicate a temporal and tissue-specific shift in the expression of NOS2 and Arg-1/-2 in vivo.

## Loss of epidermal NOS2 influence residual NOS expression and influences cardiovascular acclimation to hypoxia

To test the hypothesis that keratinocyte HIF-1α/NOS/NO influences shifts in skin temperature, systemic blood pressure and heart rates during exposure to hypoxia, we next generated mice with NOS2 deletions in keratinocytes. Deletion efficiency of epidermal NOS2 in K14cre+ mice was calculated to be greater than 98% in all mice analysed (*Figure 6B*). The deletion of epidermal NOS2

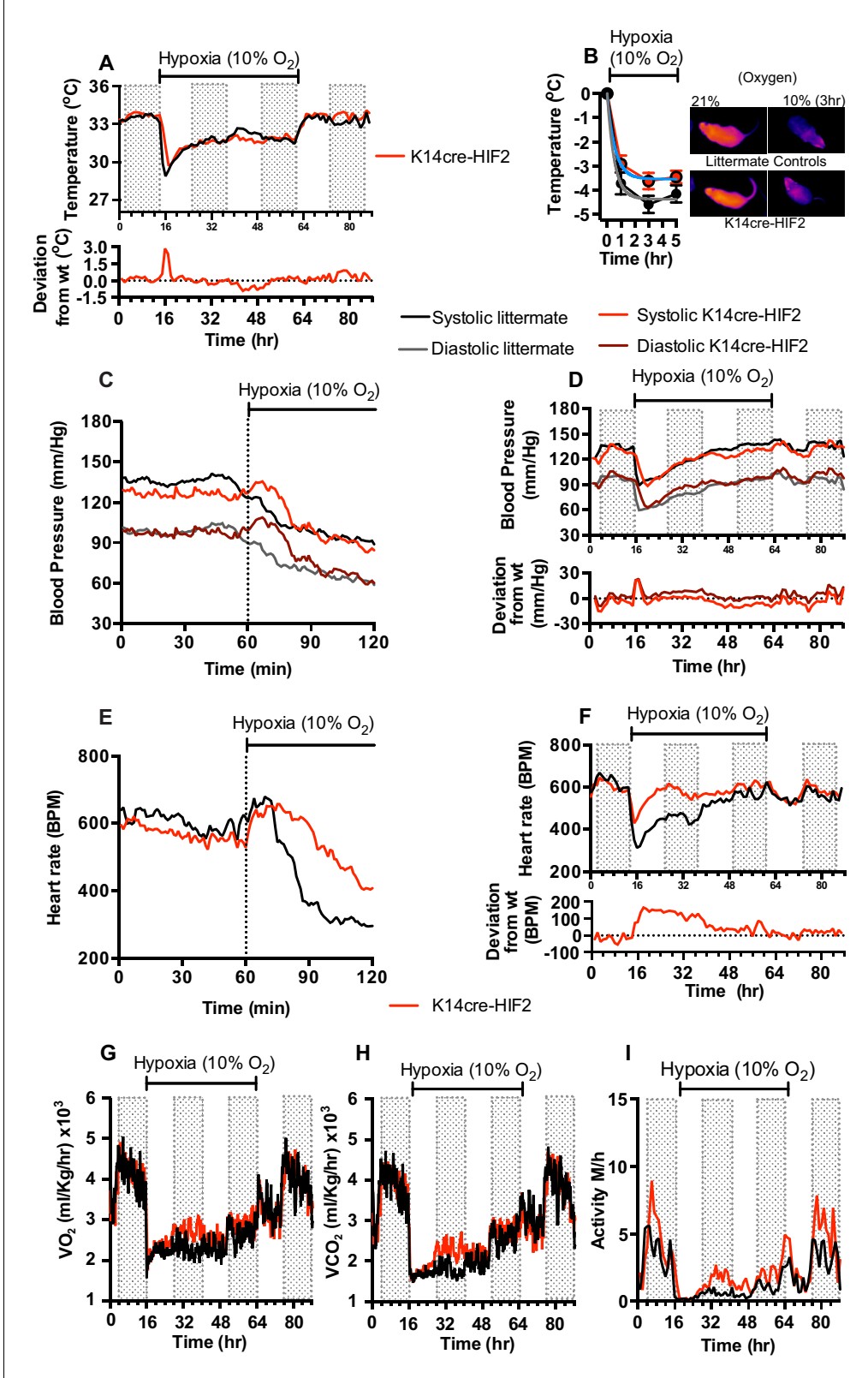

**Figure 4.** Keratinocyte deletion of HIF-2α affects cardiovascular acclimation to hypoxia. Mice were individually housed for 14 hr in normoxia before being exposed to hypoxia (10%) at the light-dark interphase for 48 hr. Mice then recovered for a further 24 hr in normoxia. (A) Skin temperature acclimation in littermate (black line, n = 7) and K14cre-HIF-2α (red line n = 7). Data shown as mean degree's Celsius (shaded area depicts dark cycle). Lower line graph shows temperature deviation of K14cre-HIF-2α from littermate. (B) Skin temperature was monitor by an infrared thermal imaging

*Figure 4 continued on next page*

*Figure 4 continued*

camera before and during hypoxia (10% O$_2$) exposure. Thermal infrared imaging suggests that K14cre-HIF-2α (n = 9) impacts skin temperature when compared to littermate (n = 5) controls during the first 5 hr of hypoxia. Photomicrographs show representative thermal images during hypoxic exposure. (C) Acute and (D) chronic analysis of systolic and diastolic blood pressure in littermate (black-line and grey-line n = 8) and K14cre-HIF-2α (dark red line and light red line n = 7) data shown as mean (mm/Hg). Lower line graph shows blood pressure deviation for both systolic (light red) and diastolic (dark red) BP. (E) Acute and (F) chronic analysis of heart rate in littermate (black line n = 6) and K14cre-HIF-2α (red line n = 6). Data shown as mean beats per minute. Lower line graph shows chronic heart rate deviation of K14cre-HIF-2α from littermate. (G) Analysis of metabolic activity during the chronic acclimation of littermate (black line, n = 6) and K14cre-HIF-2α (red line n = 7) to hypoxia. Data shown as mean oxygen consumption VO$_2$ and (H) carbon dioxide generated VCO$_2$ ml/Kg/hr. (I) Analysis of physical activity in littermate (black line n = 6) and K14cre-HIF-2α (red line n = 7) during acclimation to chronic hypoxia. Data shown as mean activity meters/hour.

DOI: https://doi.org/10.7554/eLife.28755.006

shows changes in baseline skin temperature (which is likely correlated with peripheral vascular resistance) and systemic blood pressures similar to those seen in HIF-2α mutant mice, and initial changes in heart rate similar to those seen in HIF-1α mutant mice.

As shown in *Figure 6A*, epidermal deletion of NOS2 sustains an elevated skin temperature relative to littermate controls for the first 24 hr of hypoxia, before returning to levels seen in control mice for the remainder of the recording (for statistical analysis see Figure 9K). In *Figure 6C and D*, the similarities to the HIF-2α mutant mice are evident, as the K14-NOS2 mutant animals display a profound hypotensive phenotype, with an apparent decrease in both systolic and diastolic pressures within the first 20 min of hypoxia exposure (for statistical analysis see Figure 9L). This hypotensive state is maintained throughout the hypoxic time course (*Figure 6D*). Maximal deflection of systolic pressures (−15 to −20 mmHg) occurs between 24–48 hr hypoxic exposure when compared to littermate controls (statistical analysis Figure 9M). In contrast, NOS2 mutant heart rate acclimation to hypoxia responds in a similar pattern to the HIF-1α mutants, with a profound increase in BPM 120mins after hypoxia exposure. This is also maintained throughout the first 24 hr of hypoxia exposure (*Figure 6E and F* [for statistical analysis see Figure 9N and O]).

The distinct tensive/cardiac response to hypoxia reported above may be due to a potential compensatory role of NOS1 and NOS3 expression in the skin counteracting the loss of NOS2. RT-qPCR analysis in whole skin samples identified a significant increase in the expression of both NOS1 and NOS3 in K14cre-NOS2 mice exposed to hypoxia (6 hr) when compared to littermate controls (*Figure 6G*). The compensatory roles of NOS isoform expression have been previously reported in other animal models of NOS deletion (*Colton et al., 2006*).

Interestingly, comparative analysis of VEGF-A identified no compensatory expression in K14cre-NOS2 mice (*Figure 6H*). Physical activity is greatly reduced in littermate and mutant animals during the initial 12 hr of hypoxia and slowly recovers between 12–48 hr, although it does not fully recover until the animals are restored to normoxia (*Figure 6I*).

## Loss of Arginase1 in the epidermis partially mirrors the effects seen from loss of HIF-2α

To determine if the complex shifts in skin temperature, blood pressure and heart rate during hypoxia seen in HIF-2α mutant animals were due to HIF-2α regulation of the arginase pathway, we next analysed mice with Arg1 deletions in keratinocytes. As can be seen in *Figure 7A*, and similarly to what was seen in HIF-2α mutant mice, these animals display an elevated skin temperature relative to littermate control mice during the first twelve hours of hypoxic exposure, followed by a relative decline at 24 hr (for statistical analysis see Figure 9P). In *Figure 7B* (for statistical analysis see Figure 9Q) it is apparent that there is a small increase in systolic and diastolic pressures for the first 10 min hypoxia, followed by a decrease in both systolic and diastolic pressure. This occurs in a manner similar to that of littermate control animals. The recovery of a normotensive state in K14cre-Arg1 mutant mice, is somewhat similar to that seen in littermate controls (*Figure 7C*)(for statistical analysis see Figure 9R). However, the effect on heart rate is similar in some regards to that seen in HIF-2α mutants, with an overall increase relative to the hypoxia-induced decline seen in littermate control animals (*Figure 7D*)(for statistical analysis see Figure 9S), and a more rapid return to the normal pattern of diurnal variation in heart rate (*Figure 7E*)(for statistical analysis see Figure 9T).

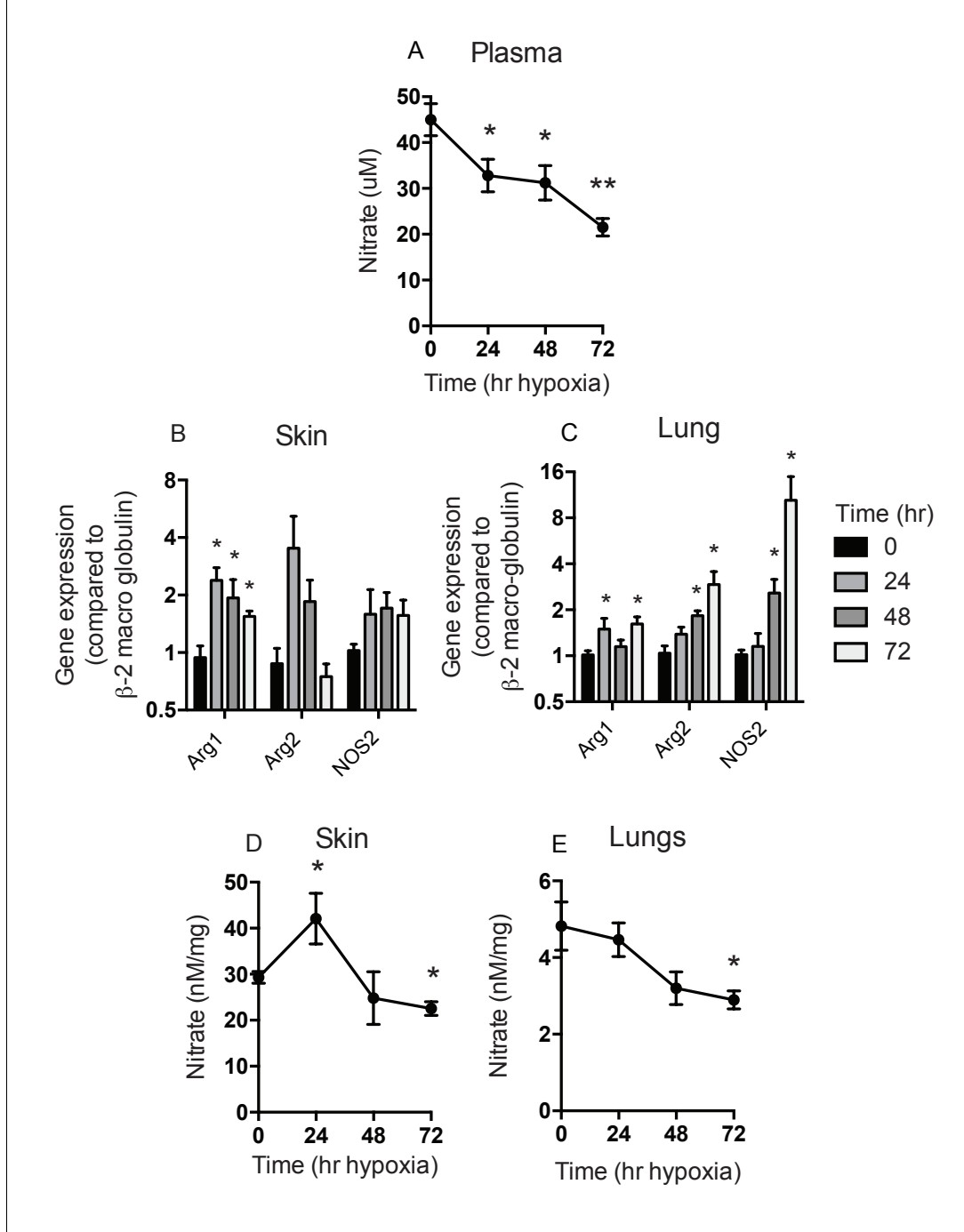

**Figure 5.** Analysis of Nitrates and NOS2–arginase gene expression. Tissues and fluids were harvested from hypoxic mice (C57/Bl6 wild-type) at the time points shown and analysed for nitrate content and the gene expression of NOS2 and arginase-1/-2. (**A**) plasma nitrate concentration shown as mean ±SEM μM (n = 8). (**B**) Skin and (**C**) lung qPCR analysis of NOS2 and arginase expression. Data shown as mean ±SEM of fold change in gene expression compared to normoxia control (β−2 macroglobulin house-keeping gene using $2^{-\Delta\Delta CT}$ method)(n = 8). (**D**) Skin and (**E**) lung nitrate concentration shown as mean ±SEM μM (n = 8). Statistical analysis using mann-Whitney test (*p≤0.05 **p≤0.005).

DOI: https://doi.org/10.7554/eLife.28755.007

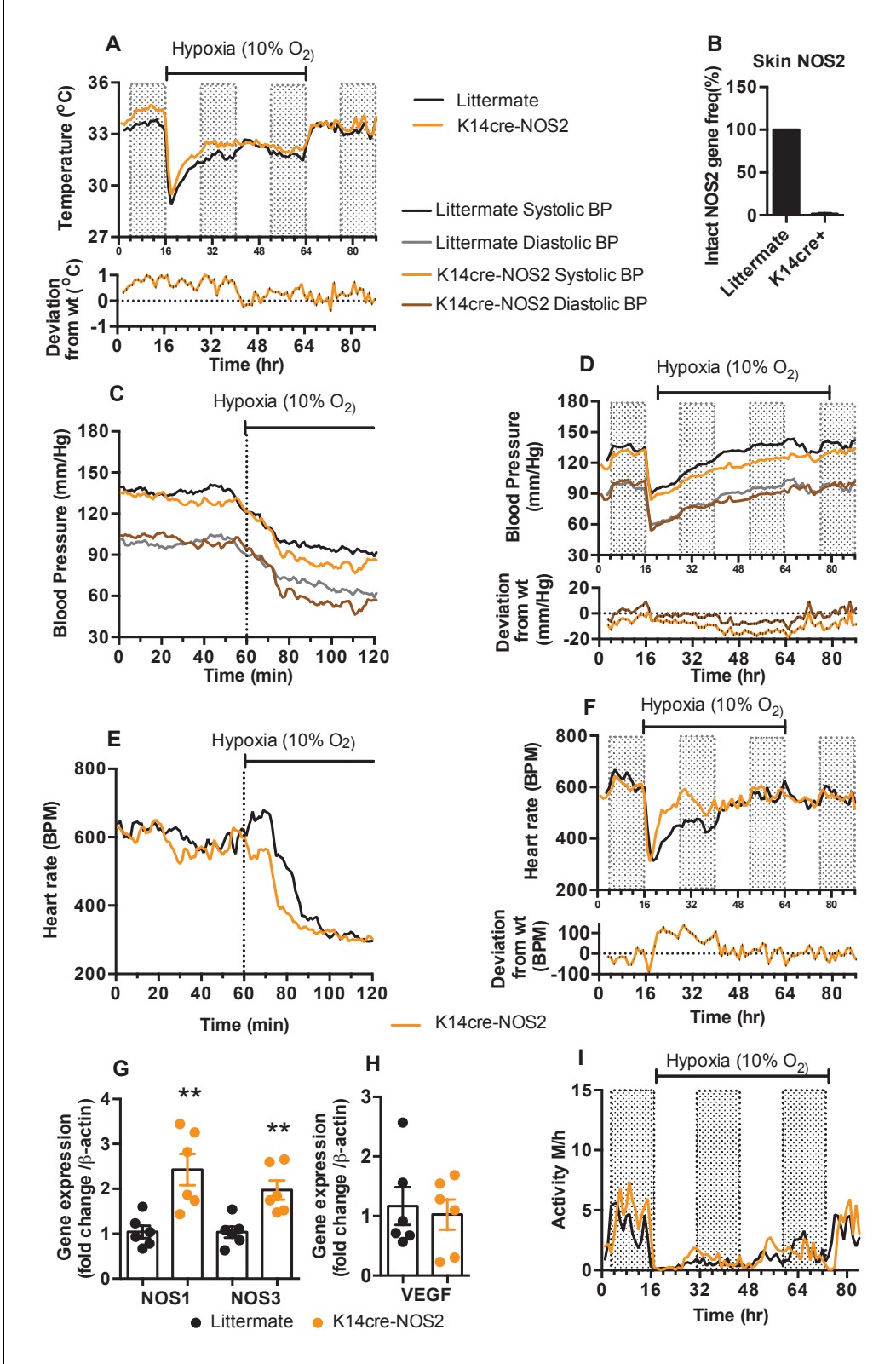

**Figure 6.** Keratinocyte deletion of NOS2 affects cardiovascular acclimation to hypoxia. Mice were individually housed for 14 hr in normoxia before being exposed to hypoxia (10%) at the light-dark interphase for 48 hr. Mice then recovered for a further 24 hr in normoxia. (A) Skin temperature acclimation in littermate (black line, n = 7) and K14cre-NOS2 (red line n = 5). Data shown as mean degree's Celsius (shaded area depicts dark cycle). (B) Deletion efficiency of NOS2 in K14cre+ mice was calculated by Taq-man PCR. (C) Acute and (D) chronic analysis of systolic and diastolic blood pressure

*Figure 6 continued on next page*

*Figure 6 continued*
in littermate (black-line and grey-line n = 8) and K14cre-NOS2 (dark red line and light red line n = 5) data shown as mean (mm/Hg). Lower line graph shows blood pressure deviation for both systolic (light red) and diastolic (dark red) BP. (E) Acute and (F) chronic analysis of heart rate in littermate (black line n = 6) and K14cre-NOS2 (red line n = 5). Data shown as mean beats per minute. Lower line graph shows chronic heart rate deviation of K14cre-NOS2 from littermate. (G and H) RT-qPCR analysis of whole skin samples from K14cre-NOS2 (n = 6) and littermate mice (n = 6) following 6 hr hypoxia (10% $O_2$) exposure. (G) Epidermal NOS2 deletion stimulates enhanced expression of NOS1 and NOS3. (H) Skin VEGF expression is not effect by NOS2. (I) Analysis of physical activity in littermate (black line n = 6) and K14cre-NOS2 (red line n = 7) during acclimation to chronic hypoxia. Data shown as mean activity meters/hour.
DOI: https://doi.org/10.7554/eLife.28755.008

Metabolic activity of the Arg-1 animals also follows a similar trajectory to that seen in HIF-2α mutant animals. Both $VO_2$ and $VCO_2$ (*Figure 7F–G*) substantially increased between 12 and 48 hr of hypoxia, and this again coincided with heightened physical activity (*Figure 7H*). In each case, the changes seen in Arg-1 mutants were less striking than those seen in HIF-2α mutants, likely due in part to the presence of Arg-2 in the skin of these animals. These data indicate that a significant aspect of the HIF-2α-mediated changes in the cardiovascular response to hypoxia may be mediated via its regulation of arginase expression.

## Ventilatory response to hypoxia and carotid body development

We next questioned whether epidermal deletion of HIFα isoforms influenced the basal or hypoxia-induced ventilatory response. Whole-body plethysmography showed that resting ventilation rates are similar in the HIFα mutant animals relative to littermate controls (*Figure 8A–B*). All mice responded normally to acute hypoxia by increasing ventilation rates. K14cre-HIF-1α, K14cre-HIF-2α, and wild type littermate control mice increased their respiratory rate for the initial 10 min of hypoxic exposure, before reducing their ventilation rates 30 min after exposure to hypoxia began (*Figure 8A–B*). Oxygen partial pressures in the blood were not significantly different from those seen in littermate animals during an acute hypoxic challenge (*Figure 8C*).

Likewise, histological examination of the carotid bodies showed that there were no discernable differences in the size or number of oxygen sensing cells in the mutant animals when compared to those of littermate control mice (*Figure 8D–F*). *Figure 8G* shows representative photomicrographs of TH[+] cells in the carotid bifurcation in littermates and K14cre-HIF-2α mice. Please see *Figure 9A-T* for graphic depictions of the statistical analyses characterizing these results.

## Discussion

We believe that this is the first detailed investigation of cardiovascular acclimatization to hypoxia in mice over the time frames used here. Although there have been a number of previous studies examining cardiovascular hypoxic response, these have either ended after approximately ten minutes of observation and used anaesthesia, or have involved long term exposures, with monitoring after days or weeks of conditioning (*Campen et al., 2004*; *Campen et al., 2005*). Here, we monitored animals breathing 10, 12, 15, and 21% $FiO_2$ for 48 hr, followed by a 24 hr recovery period. We found that mice exhibited a well-documented elevation of heart rate and blood pressure for the first ten minutes of exposure to hypoxia(*Campen et al., 2004*; *Yu et al., 1999*; *Dematteis et al., 2008*; *Pearson et al., 2007*), but then, experience a dramatic drop in both parameters. The maximal negative deflection in heart rate and blood pressure was observed at 3 hr post-onset of hypoxic exposure, which coincided with increased peripheral vascular resistance, and very low physical activity. Cardiovascular parameters were approximately restored to normoxic levels over the next 45 hr. These paths of decline and recovery are striking adaptive responses to hypoxia, and they appear to be graded relative to the degree of hypoxic challenge.

The processes controlling cardiovascular responses to hypoxia are complex, and include chemo-receptors and baroreceptors as part of an autonomic nervous system reflex. These neurally-mediated responses also interact with, and may frequently mask, local tissue effects of hypoxia on cardiovascular responses, making it difficult to dissociate extrinsic and local tissue responses (*Bärtsch and Gibbs, 2007*).

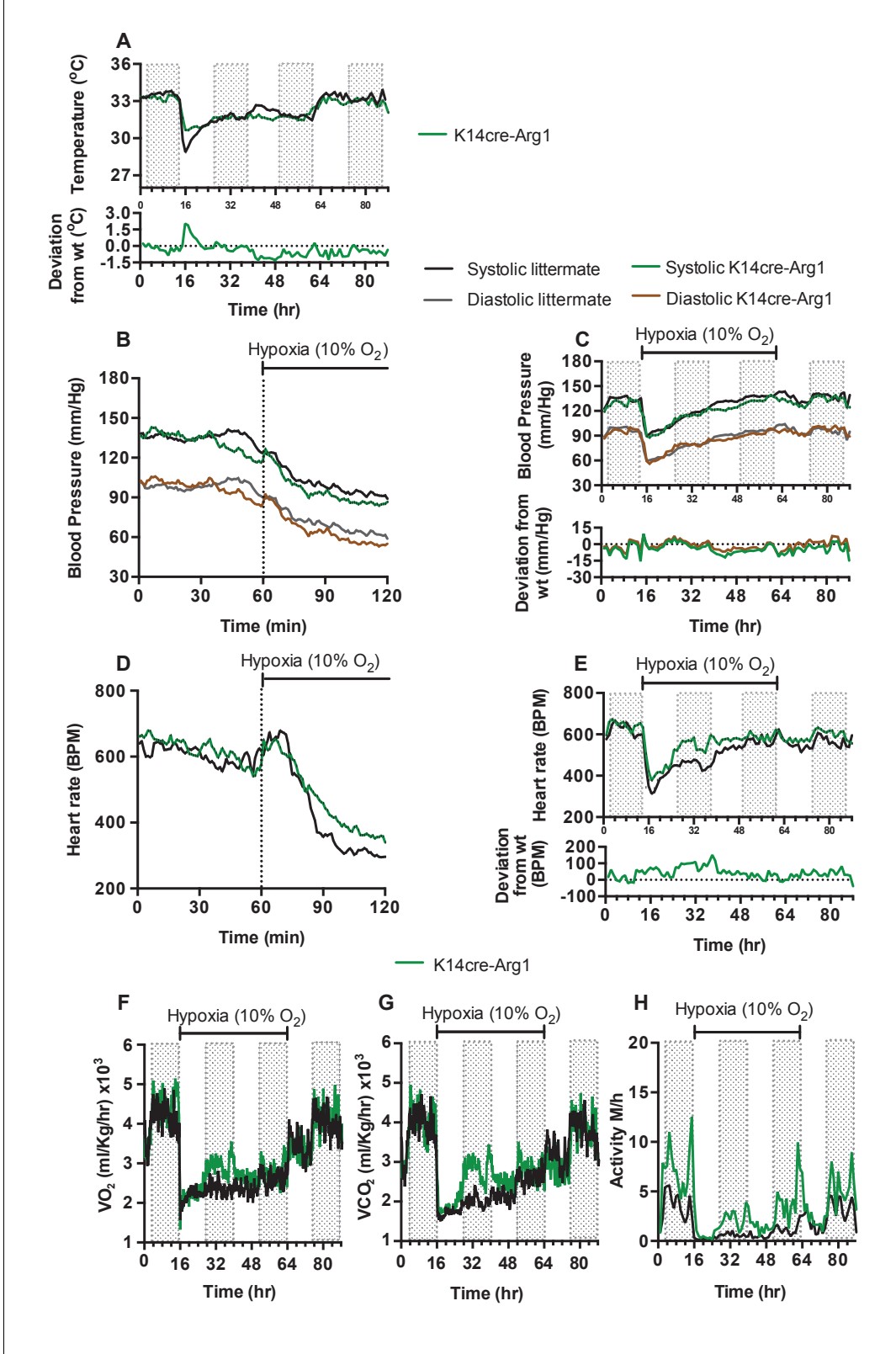

**Figure 7.** Keratinocyte deletion of arginase-1 affects cardiovascular acclimation to hypoxia. Mice were individually housed for 14 hr in normoxia before being exposed to hypoxia (10%) at the light-dark interphase for 48 hr. Mice then recovered for a further 24 hr in normoxia. (A) Skin temperature acclimation in littermate (black line, n = 7) and K14cre-arginase-1 (green line n = 6). Data shown as mean degree's Celsius. Lower line graph shows temperature deviation of K14cre-arg-1 from littermate. (B) Acute and (C) chronic analysis of systolic and diastolic blood pressure in littermate (black-line

*Figure 7 continued on next page*

*Figure 7 continued*

and grey-line n = 6) and K14cre-arg-1 (dark green line and brown line n = 6) respectively. Data shown as mean (mm/Hg). Lower line graph shows blood pressure deviation for K14cre-arg1 compared to littermate for both chronic systolic (green line) and diastolic (brown line) BP. (**D**) Acute and (**E**) chronic analysis of heart rate in littermate (black line n = 6) and K14cre-arg-1 (green line n = 6). Data shown as mean beats per minute. Lower line graph shows chronic heart rate deviation of K14cre-arg-1 from littermate. (**F**) Analysis of metabolic activity during the chronic acclimation of littermate (black line, n = 6) and K14cre-arg-1 (green line n = 6) to hypoxia. Data shown as mean oxygen consumption $VO_2$ and (**G**) carbon dioxide generated $VCO_2$ ml/Kg/hr. (**H**) Analysis of physical activity in littermate (black line n = 6) and K14cre-arg-1 (green line n = 6) during acclimation to chronic hypoxia. Data shown as mean activity meters/hour.

DOI: https://doi.org/10.7554/eLife.28755.009

We have previously documented that a balance in keratinocyte HIF-1α and HIF-2α expression modulates peripheral vascular resistance (*Boutin et al., 2008*), and that this directly affects murine systemic blood pressure; it is also correlated with the tonic tensive state in mildly hypertensive humans (*Cowburn et al., 2013*). Here we show for the first time that keratinocyte HIF-1α and HIF-2α expression affects the immediate, acute and chronic phases of the cardiovascular response to hypoxia.

Hypoxia immediately heightens sympathetic tone in the skin to increase peripheral vascular resistance (*Simmons et al., 2007*; *Chalmers and Korner, 1966*; *Kollai, 1983*). Hypoxia also increases nitric oxide release from keratinocytes to stimulate vasodilation and act as a neuromodulator of sympathetic activity, dampening vasoconstriction (*Ramchandra et al., 2005*; *Sartori et al., 2005*; *Hirooka et al., 2011*). The opposing roles of keratinocyte HIF-1α and HIF-2α on nitric oxide synthesis (*Cowburn et al., 2013*) would suggest divergent effects on sympathetic activity and blood flow in this highly vascular tissue.

Epidermal HIF-1α is shown here to have a direct effect on peripheral vascular resistance and the tonic tensive status of the mouse. The opposite cardiovascular relationship is apparent in the HIF-2α mutant mouse. In the case of the HIF-2α mouse, however, there is a significant increase in heart rate, likely to maintain systolic blood pressure. In this mutant, there appears to be a transition/acclimation checkpoint following 24 hr in hypoxia. The heart rates of both HIF-2α mutant and littermate mice converge at this point, and systolic pressure of the HIF-2α mutant drops below that of controls. This suggests a sustained decrease in the vascular resistance of the skin. Epidermal deletion of arginase-1 in this hypoxia acclimation model phenocopied several characteristics of the K14cre-HIF-2α mutant mouse. There is an overall lessening of the effect, however, indicating that other genes controlled by HIF-2α are likely involved in the hypoxic acclimation process.

These alterations in peripheral vascular blood flow also appear to affect whole body metabolism and physical activity during the acclimation phase to hypoxia. Although all four epidermal mutant mice described here demonstrated a similar increase in oxygen consumption during this acclimation phase, only K14cre-HIF-2α and K14cre-arginase-1 mice demonstrated some preservation of physical activity in comparison to wild-type animals. This is intriguing, as it indicates that the response of the peripheral tissue can modulate the animals overall behaviour during environmental stress.

Interestingly, the data presented here are consistent with cardiopulmonary investigation of patients with germline, heterozygous gain-of-function mutations in HIF-2α, where baseline heart rate and cardiac responses to moderate hypoxia were shown to be higher in the HIF-2α-gain of function patients when compared to control subjects, suggesting an increase in cardiac sympathetic tone (*Formenti et al., 2011*).

In summary, we present here the first dissection of the murine cardiovascular response to hypoxia, and we present evidence that this response is highly influenced by peripheral vascular responses. This indicates that peripheral tissues, and their vascular beds, form a complex physiological network of oxygen-responsive tissues that influence, and are not just influenced by, central regulators of cardiovascular function. This interaction amongst tissues indicates that cardiovascular responses to hypoxia are likely a summation of a range of tissue responses, and not solely dictated by central mediators of cardiovascular activity.

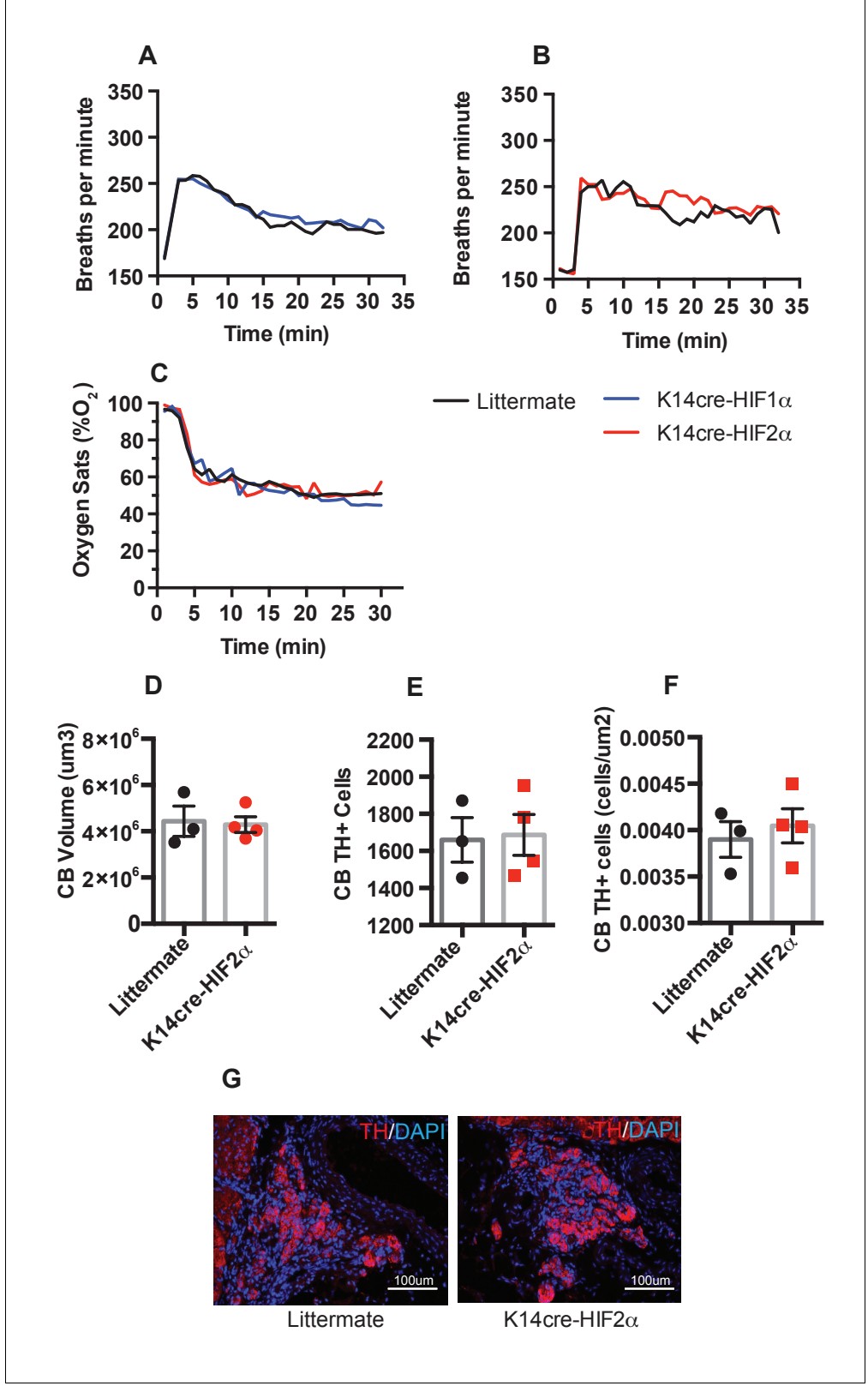

**Figure 8.** Pulmonary respiratory response to acute hypoxia. (**A and B**) Ventilation rate in response to acute hypoxia was determined by whole-body plethysmography. Resting/normoxia ventilation was determined 60 min before acute hypoxic stimulus. Data shown as mean breaths/minute (HIF-1α n = 6, HIF-2α n = 4) (**C**) Percentage arterial oxygen saturation was recorded during the acute hypoxic challenge. Data recorded at 5 s interval mean

*Figure 8 continued on next page*

*Figure 8 continued*

(HIF-1α n = 6, HIF-2α n = 4). Keratinocyte HIF-2α deletion does not influence carotid body development. Quantification of (D) carotid body volume, (E) CB TH⁺ cells (F) CB TH⁺ cells per area tissue, (G) representative photomicrographs of TH⁺ cells in the carotid bifurcation in littermates and K14cre-HIF-2α mice.
DOI: https://doi.org/10.7554/eLife.28755.010

## Materials and methods

### Animals

All animals were housed in an association and accreditation of laboratory animal care international-approved facility. All protocols and surgical procedures were approved by the UK Home Office and the University of Cambridge Animal Welfare Ethical Review Board (AWERB) under project license 80/2565 from the UK Home Office.

Targeted deletion of *Hif1a* (HIF-1α), *Epas1* (HIF-2α), *Nos2* (NOS2)and *Arg1* (arginase-1) in keratinocytes was accomplished by crossing mice (in a C57/Bl6J background ascertained by extensive backcrossing and SNP analysis) homozygous for the floxed allele in HIF-1α, HIF-2α, NOS2 or Arginase-1 into a background of Cre recombinase expression driven by the K14 promoter, which is specific to cells of the keratinocyte linage. The NOS2 mouse was generated by Ozgene (Perth, Aus) in C57Bl6 embryonic stem cells via homologous recombination, and uses a placement of the loxP sites flanking exons 3. Cre recombinase-mediated deletion of the loxP-flanked exon introduces a translational frameshift, rendering downstream exons non-functional.

### Radio-telemetry implantation

All radio-telemetry hardware and software was purchase from Data Science International. (St Paul, MN, USA). All procedures regarding preparation of the transmitter are carried out in aseptic conditions. The regulated procedure describing the aseptic implantation of the radio-telemetry device has previously been described (*Cesarovic et al., 2011*). Data acquisition only commenced following the complete recovery of the animal from the regulated surgical procedure (at least 10 days). All baseline telemetry data was collected over a 96 hr period in a designated quiet room to ensure accurate and repeatable results. Radio-telemetry/hypoxia challenge was conducted in combination with Oxymax Lab animal monitoring system (CLAMS)(Columbus Instruments, Columbus, OH, USA). Mice were placed in environmental chambers and allowed to acclimate for 24 hr before the oxygen content of the flow gas was reduced to 10%. The mice were continuously monitored for the next 48 hr before being returned to normal atmospheric oxygen for a further 24 hr.

### Carotid body histology

Carotid body histology was performed as previously reported (*Macías et al., 2014*). Briefly, carotid bifurcations were dissected, fixed for 2 hr with 4% paraformaldehyde (Santa Cruz) and cryopreserved with 30% sucrose in PBS. 10 μm-thick cryosections were obtained (Bright cryostat) and tyrosine hydroxylase (TH) positive cells were detected by immunofluorescence using rabbit anti-TH antibody (Novus bioscience ref: 300–109) and goat anti-rabbit Alexa568 antibody (Life technologies ref: A11036). Carotid body volume and cell numbers were quantified on microscope images (Leica DM-RB) using ImageJ software.

### Whole body unrestrained plethysmography

A single chamber plethysmograph (Data Science) was used in conjunction with a pressure transducer. This utilises the barometric analysis technique that compares the pressure difference between the animal chamber and a reference chamber to measure airway physiological parameters. Unanesthetized mice were randomly placed into the plethysmograph and allowed to acclimate. Baseline averages of breathing frequency, tidal volume, inspiration and expiration volumes/times were recorded. Once acclimated to the chamber, the composition of the flow gas was switched from 21% $O_2$ to 10% $O_2$ using a PEGAS mixer (Columbus instruments). The mice were housed in the reduced oxygen environment for 60 min before being returned to atmospheric oxygen.

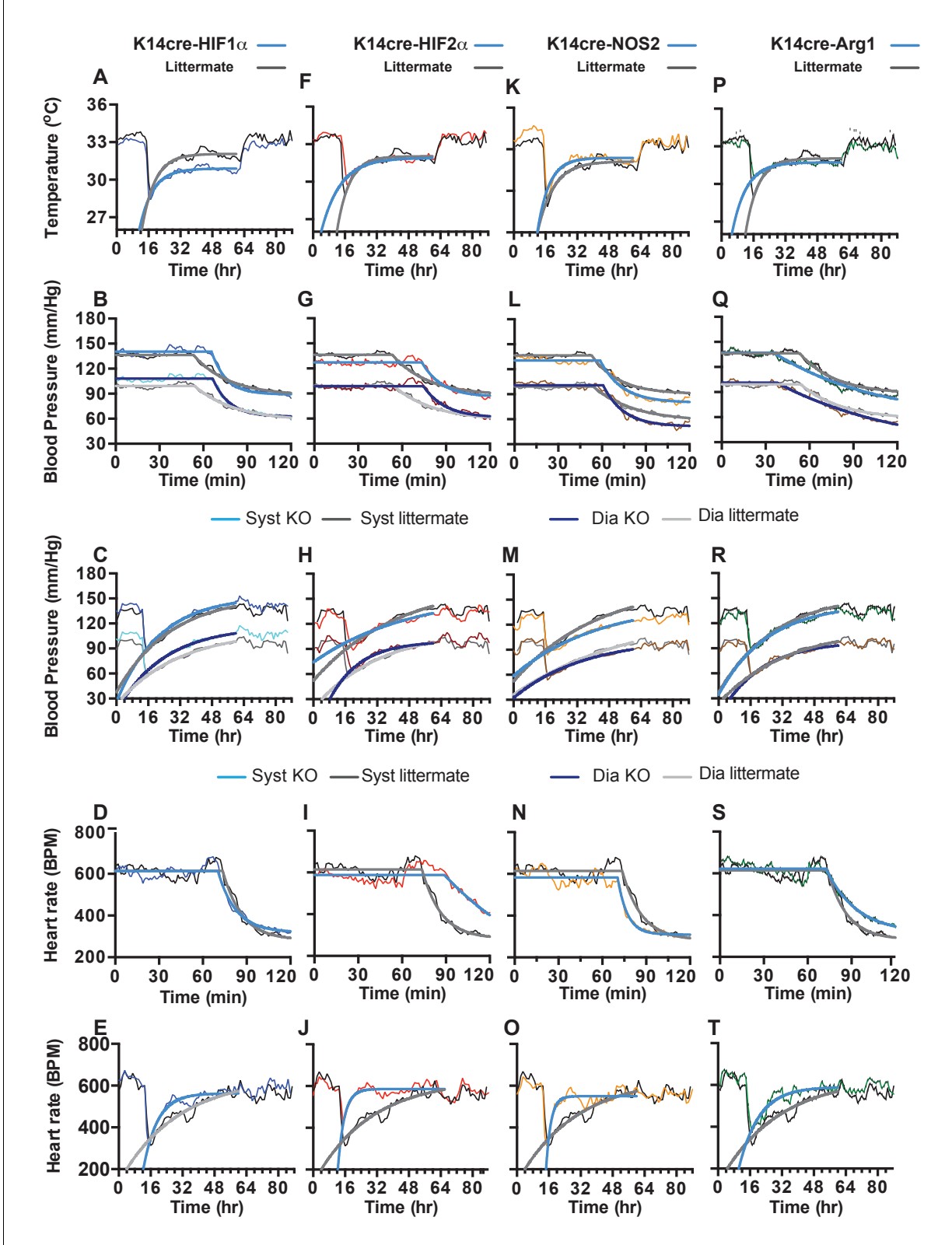

**Figure 9.** Statistical analysis of K14cre-HIF-1α, HIF-2α, NOS2 and Arg-1 compared to littermate control for skin temperature. (A, F, K, P), systolic and diastolic blood pressure following acute hypoxia (B, G, L, Q), systolic and diastolic blood pressure following chronic hypoxia (C, H, M, R), heart rate following acute hypoxia (D, I, N, S) and heart rate following chronic hypoxia (E, J, O, T). Non-linear regression modelling was undertaken, using a least

*Figure 9 continued on next page*

*Figure 9 continued*

squares method, with Akaike's Informative Criteria (AICc) used to determine whether the data could best be represented by a single model, or separate ones. The probablility of the data being best represented by two different models was >99.99% in all cases.

DOI: https://doi.org/10.7554/eLife.28755.011

### Nitrite/nitrate analysis

Blood samples were centrifuged to separate plasma and were passed through a column with a 10 kDa cut-off filter. All samples were analysed for total $NO_{(X)}$ content using a NOA 280i (Siever, GE Healthcare) according to the manufacturers instructions.

### Isolation and culture of primary keratinocytes

The preparation of primary keratinocytes from adult mouse tail skin is comprehensively described by *Lichti et al. (2008)*.

### RNA/DNA isolation and qPCR

Total RNA was isolated from skin using TRI-reagent (Sigma), followed by RNA clean-up and DNase digest using RNeasy column kits (Qiagen, Manchester, UK). First-strand synthesis was performed with 1 µg of total RNA using a high-capacity cDNA kit (Applied Biosystems, Paisley, UK) according to the manufacturer's instructions. Relative gene expression was determined by qPCR (ABI system, Applied Biosystems) and amplified in Sybr-green master mix (Roche) using relevant primers from Qiagen. Deletion efficiency was characterised for NOS2 in the K14cre+ mouse using DNA isolated from skin samples using TRIzol/DNeasy columns (Qiagen, Manchester. UK). The PCR primers and Taqman probe were designed in-house and synthesised by Sigma (Gillingham UK) fwd 5'-TCCAGAA TCCCTGGACAAG rev 5'-TGGTGAAGAGTGTCATGCAA, probe 5'-FAM-TGTGACATCGACCCG TCCACA.

### Metabolic analysis

Energy expenditure of the K14cre mice and their littermate controls was measured using the Columbus Instruments Oxymax system according to the manufacturers instructions (*Cowburn et al., 2013*).

### Mouse skin temperature analysis

Surface temperatures were measured with a FLIR Thermovision A20 thermal infrared camera, and image data were analyzed using FLIR image analysis software

### Statistical methods

Baseline radiotelemetry parameters in *Figure 1* were analysed by determining the area under curved for each mouse per group and then performing a mann-Whitney test to determine the statistical difference across groups compared to littermate. Mann-Whitney test was used to analyse RT-qPCR gene expression and plasma/tissue nitrate data in *Figure 5*.

Non-linear regression modelling, using a least squares method, was used to model the radiotelemetry data in *Figures 3*, *4*, *6* and *7* Akaike's informative Criteria (AICc) was used to compare models, and determine whether the data were best represented by a single model, or by separate ones. Analysis was undertaken using GraphPad Prism Version 7.0 c for Mac OS.

## Additional information

### Funding

| Funder | Author |
| --- | --- |
| Wellcome | Randall S Johnson |

The funders had no role in study design, data collection and interpretation, or the decision to submit the work for publication.

## Author contributions
Andrew S Cowburn, Conceptualization, Formal analysis, Supervision, Validation, Investigation, Methodology, Writing—original draft, Writing—review and editing; David Macias, Investigation, Methodology, Writing—review and editing; Charlotte Summers, Formal analysis, Methodology, Writing—review and editing; Edwin R Chilvers, Validation, Methodology, Writing—review and editing; Randall S Johnson, Conceptualization, Resources, Data curation, Formal analysis, Supervision, Funding acquisition, Validation, Investigation, Visualization, Methodology, Writing—original draft, Project administration, Writing—review and editing

## Author ORCIDs
Edwin R Chilvers (iD) https://orcid.org/0000-0002-4230-9677
Randall S Johnson (iD) http://orcid.org/0000-0002-4084-6639

## Ethics
Animal experimentation: All animals were housed and all experiments carried out according to approved protocols and under the guidance and approval of the UK Home Office, under the auspices of an approved Program Project License, and with ethical approval of the University of Cambridge Animal Welfare Ethical Review Board under project license number 80/2565.

## Decision letter and Author response
Decision letter https://doi.org/10.7554/eLife.28755.015
Author response https://doi.org/10.7554/eLife.28755.016

# Additional files

## Supplementary files
• Supplementary file 1. Spreadsheet of means and standard deviations for all points described in the graphs of *Figures 1–9*.
DOI: https://doi.org/10.7554/eLife.28755.012

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
