## [Decision Letter]

Thank you for submitting your article "Cardiovascular adaptation to hypoxia: the role of peripheral resistance" for consideration by *eLife*. Your article has been favorably evaluated by Didier Stainier (Senior Editor) and three reviewers, one of whom, Jan-Marino Ramirez (Reviewer #1), is a member of our Board of Reviewing Editors.

The reviewers have discussed the reviews with one another and the Reviewing Editor has drafted this decision to help you prepare a revised submission.

Summary:

This is an important and novel study by Cowburn et al. that significantly contributes to our understanding of long-term adaptation to hypoxia. The authors provide evidence that keratocytes play an important role in the midrange adaptation to chronic hypoxia. They specifically demonstrate that the cutaneous HIF1-HIF2-axis plays a key role in cardiovascular responses upon exposure of mice to low oxygen environments. The authors argue that HIF1α promotes via NOS2-dependent NO production local cutaneous vasodilatation. In contrast, HIF2α antagonizes this response by promoting local skin Arginase expression and thereby diminishes Arginine, which is a critical substrate for NOS2.

The main concern raised by the reviewers centered on the statistical analysis and some of your conclusions. As you will see, we provide specific recommendations to address this concern.

Essential revisions:

1) The study reports several observations with a low "n", and in these cases the numbers should be increased. As an example, in the last paragraph of the subsection “Loss of HIFα isoforms in the epidermis differentially affects blood pressure and skin temperature across diurnal cycles” the authors state that there was a tendency towards a temperature effect in HIF2α mutants. This statement is based on an "n" of only 4. Clearly this finding is "underpowered" and more n's are needed to come to a meaningful conclusion.

2) The Figure 8 summarizes the "statistical analysis". Several of the findings really don't look significantly different. E.g. I don't see how differences in the heart rate changes as shown in Figure 8 or 8N should be highly significant. The same applies to Figure 8; the differences seem quite subtle. This is consistent with our general concern (see next comments), that the statistical analysis was not always appropriate.

3) The authors stated that two-way ANOVO with Dunnett's test was used to compare K14cre-mutants to wild-type controls under baseline condition in Figure 1.

While two-way ANOVA is a good option for testing statistical difference when combining strains with treatments (baseline, hypoxia, recovery). However, given the fluctuation of data I believe there is a significant interaction between two independent variables. Thus, we urge the authors to consider additional tests.

4) The use of un-paired t-test for analyzing gene expression is not appropriate since the values tested have been normalized, and the normalization could violate one or more of the underlying assumptions for the test. A non-parametric test should be used instead.

5) The Akaike information criterion (AIC) measures the relative quality of models but does not test a null hypothesis. Thus, the p values in Figure 8 are misleading and should be removed from this figure.

6) The authors did not mention what statistics methods were used for testing differences in cardiovascular radio-telemetry parameters among strains during hypoxia and recovery. This information should be provided.

7) As already mentioned above, in several instances the sample size of each group is too small (i.e., n=2-6) and makes the results questionable. Variation in each group should be reported in a certain form (e.g., text, table or figure), and additional n's should be provided.

8) Please clarify the following issue: Peripheral resistance has been considered a major contributor to diastolic pressure. If HIF-2α is important in regulating peripheral resistance as proposed by the authors, how do you interpret that effect of keratinocyte deletion of HIF-2α is more profound in systolic than diastolic pressure, as shown in Figure 4 hr, 48-64 hr)? Indeed, diastolic pressure during 0-16 hr and 48-64 hr (Figure 4) seems not to be different in HIF-2α KO from WT.

9) NO production by NOS requires molecular O_2_. How do you interpret the increased NO production in the skin during prolonged hypoxia (Figure 5)? Please also explain why the biochemical measurements were obtained from tissues of mice treated with 72 hrs hypoxic exposure, whereas physiological measurements were made in mice treated with 48 hrs hypoxia?

10) The authors argue that HIF1α affects peripheral resistance. This statement is based on quantification of skin body temperature as surrogate parameter (subsection “Cardiovascular responses to 48 hours of hypoxia: brief initial hypertension and tachycardia, followed by hypotension and bradycardia”, third paragraph). Alternatively, HIF1-HIF2 might impact on skin temperature by affecting mitochondrial energy production (e.g. Thomas et al., 2017, AJPRP). The authors might want to consider backing up these findings with experiments that directly demonstrate effects of HIF1-HIF2 axis on skin tissue perfusion.

11) As far as we understand the authors used WT mice as controls for their conditional skin specific k.o. mice. I wonder whether the authors performed controls that demonstrate that littermates of their conditional Hif1, Hif2a and Arg1 mice show similar responses as did the WT BL/6 mice?

12) The impaired suppression of heart rate upon hypoxic exposure in K14-HIF2 k.o. is impressive (Figure 4). The authors argue that K14-Arg1 k.o. display similar findings (Figure 6). However, the figures are rather different. Therefore, we don't agree with the authors' statement that "the effect on heart rate is similar". These data rather suggest that K14-HIF2 differentially affect heart rate, which is not linked to Arg1 expression. In K14-HIF2 HIF1 might compensate for losses of HIF2. Along this line, there is some evidence that in addition to HIF2 (as shown by the authors in previous studies) HIF1 might regulate Arg1 as well (Colegio et al., Nature, 2014). Overall, we are not sure whether all the effects observed in K14-HIF2 mice are due to diminished Arg1 expression. Please clarify.

13) The authors demonstrated earlier that skin HIF-1 is required for renal EPO production (Cell 2008). Therefore, it remains unclear whether the cardiovascular responses observed in the K14-Hif1 situation are due to local regulation of tissue perfusion (as suggested in this paper) or affected by alterations in blood viscosity and blood oxygenation (due to suppressed EPO production). Experiments that dissect the contribution of EPO vs. local effects of HIF1 would help to clarify this issue (e.g. injection of rm-EPO in K14-Hif1 mice).

14) Please better emphasize the novelty (specifically the mechanistic insights) of the present study, compared to the cardiovascular characteristics of the mutant mice reported previously by the same group (PNAS, 2013; (Cowburn et al., 2013).

15) Cardiovascular control is a highly integrative process and peripheral resistance is one of major determinates of blood pressure. How do you integrate your finding to the overall control of blood pressure?

16) The authors need to explain whether the delayed cardiovascular responses in K14Cre-HIF-1α mice are due to higher baseline blood pressure. Likewise, changes in heart rate responses in K14Cre-HIF-2α mice might as well be due to baseline bradycardia.

---

## [Author Response]

Essential revisions:1) The study reports several observations with a low "n", and in these cases the numbers should be increased. As an example, in the last paragraph of the subsection “Loss of HIFα isoforms in the epidermis differentially affects blood pressure and skin temperature across diurnal cycles” the authors state that there was a tendency towards a temperature effect in HIF2α mutants. This statement is based on an "n" of only 4. Clearly this finding is "underpowered" and more n's are needed to come to a meaningful conclusion.

We appreciate the concern regarding the sample number in this study, and have spent the last two months adding to the animal number studied to alleviate this concern. We have now increased the experimental n by implanting radio-telemetry probes and carrying out studies in an additional 16 mice in the existing groups, as well as adding a new experimental group of animals with keratinocyte-specific deletions in the NOS2 gene. This has increased baseline recording to between 7-10 mice for K14cre-HIF1α, K14cre-HIF2α and all littermate controls. Additional mice were also exposed to the chronic hypoxia protocol, and that data has now been incorporated into our analyses. New data has been incorporated into Figure 1, Figure 2, Figure 3, Figure 4, Figure 5, Figure 7 and 9.

Statistical analyses were completed and have now been included in a new Figure 9.

2) The Figure 8 summarizes the "statistical analysis". Several of the findings really don't look significantly different. E.g. I don't see how differences in the heart rate changes as shown in Figure 8 or 8N should be highly significant. The same applies to Figure 8; the differences seem quite subtle. This is consistent with our general concern (see next comments), that the statistical analysis was not always appropriate.

We have now undertaken a new series of analyses, which are depicted in Figure 9. We now use the Akaike Information Criterion (AIC) to determine the relative quality of the models. This analysis is not hypothesis driven, and so does not produce a p value.

3) The authors stated that two-way ANOVO with Dunnett's test was used to compare K14cre-mutants to wild-type controls under baseline condition in Figure 1.While two-way ANOVA is a good option for testing statistical difference when combining strains with treatments (baseline, hypoxia, recovery). However, given the fluctuation of data I believe there is a significant interaction between two independent variables. Thus, we urge the authors to consider additional tests.

We appreciate the reviewer comments regarding the statistical analyses of Figure 1. We have now increased the n value for this figure and analysed the curves by determining area under curve of each mouse per group and used a non-parametric Mann-Whitney analysis to compare mutant animal groups to littermate controls.

4) The use of un-paired t-test for analyzing gene expression is not appropriate since the values tested have been normalized, and the normalization could violate one or more of the underlying assumptions for the test. A non-parametric test should be used instead.

We have now increased the n value of all components. This has allowed the use of a non-parametric Mann-Whitney test to determine significance.

5) The Akaike information criterion (AIC) measures the relative quality of models but does not test a null hypothesis. Thus, the p values in Figure 8 are misleading and should be removed from this figure.

We agree. The p values have now been removed.

6) The authors did not mention what statistics methods were used for testing differences in cardiovascular radio-telemetry parameters among strains during hypoxia and recovery. This information should be provided.

We would like to draw the attention of the reviewer to the paragraph in the Materials and methods section stating that: “Non-linear regression modelling, using a least squares method, was used to model the radio-telemetry data in Figure 3, Figure 4, Figure 6 and Figure 7 Akaike Information Criteria (AIC) was used to compare models, and determine whether the data were best represented by a single model, or by separate ones. Analysis was undertaken using GraphPad Prism Version 7.0c for Mac OS.”

7) As already mentioned above, in several instances the sample size of each group is too small (i.e., n=2-6) and makes the results questionable. Variation in each group should be reported in a certain form (e.g., text, table or figure), and additional n's should be provided.

The sample size of animal groups has now been increased. We have implanted an additional 16 radio-telemetry probes for the acute/chronic cardiovascular acclimation to hypoxia, and increased each component of Figure 5 (nitrate analysis and gene expression) with an additional 5 mice. We understand the reviewers concerns with regards to the visualisation of the telemetry data set, shown as mean values. Our original aim was to provide the reader with a clean line graph for clear interpretation. We now provide an excel spreadsheet containing mean and standard error of mean across the time course for all telemetry figures so as to provide all information necessary for readers to interpret our findings.

8) Please clarify the following issue: Peripheral resistance has been considered a major contributor to diastolic pressure. If HIF-2α is important in regulating peripheral resistance as proposed by the authors, how do you interpret that effect of keratinocyte deletion of HIF-2α is more profound in systolic than diastolic pressure, as shown in Figure 4 hr, 48-64 hr)? Indeed, diastolic pressure during 0-16 hr and 48-64 hr (Figure 4) seems not to be different in HIF-2α KO from WT.

The reviewer makes an important point here. We hope it is apparent that in this manuscript we are just beginning an analysis of the complexity of tissue-specific feedback on regulation of cardiovascular response. The most straightforward way for this to occur is via localized changes in peripheral resistance; but clearly, there are other potential means for peripheral tissues to alter cardiovascular response, and the HIF system regulates many different genes that can accomplish this. As changes in peripheral resistance do occur in these models, along with changes in diastolic pressure, we have focused much of our discussion on those changes, but have touched on the other multiple factors that will need to be considered in interpreting and ultimately expanding on these findings.

9) NO production by NOS requires molecular O_2_. How do you interpret the increased NO production in the skin during prolonged hypoxia (Figure 5)? Please also explain why the biochemical measurements were obtained from tissues of mice treated with 72 hrs hypoxic exposure, whereas physiological measurements were made in mice treated with 48 hrs hypoxia?

The explanation for how NO generation can increase when oxygen levels drop is simply that this is due to increases in NO synthase levels, coupled to increases in L-arginine availability; these are able to compensate for lowered oxygen availability. It should be noted that there are still very significant levels of oxygen available in tissues and cells at environmental oxygen levels of 10%.

We have now increased the n values for Figure 5 and included data for 48hrs hypoxia.

10) The authors argue that HIF1α affects peripheral resistance. This statement is based on quantification of skin body temperature as surrogate parameter (subsection “Cardiovascular responses to 48 hours of hypoxia: brief initial hypertension and tachycardia, followed by hypotension and bradycardia”, third paragraph). Alternatively, HIF1-HIF2 might impact on skin temperature by affecting mitochondrial energy production (e.g. Thomas et al., 2017, AJPRP). The authors might want to consider backing up these findings with experiments that directly demonstrate effects of HIF1-HIF2 axis on skin tissue perfusion.

We appreciate the reviewers concerns, and have performed two additional sets of experiments in response. First, we have analysed skin surface temperature using a Flir infra-red thermal imaging camera. These measurements were taken from littermate controls and K14cre-HIF1α and K14cre-HIF2α mice in normoxia, followed by repeat measures at 1-3-5hrs under hypoxia (10% O_2_) We have performed a non-linear regression analysis and asked whether a single line would fit both data sets or whether a separate line would be required for each data set. However this method focuses on surface skin temperature. We would also like to make available laser Doppler data for the reviewers only. We have spent a considerable amount of time fine tuning anaesthetic delivery and thermoregulatory parameters in order to determine peripheral vascular response to hypoxia. Our protocol consists of anaesthetising a mouse with isoflurane 2% at 2L/min oxygen, after which this is quickly reduced to 1% isoflurance on a nose cone. The mouse is kept thermoneutral on a lightly heated blanket and temperature monitored. The laser Doppler starts recording hind-limb blood flow within 3 minutes of anaesthetic induction. The Doppler takes very high definition images over a 1 minute recording time.

Initially, 3 images were recorded, before the composition of the anaesthetic carrier gas was altered to 10% O_2_ (Author response image 1). Images were recorded for an additional 32 minutes. Peripheral perfusion was analysed frame by frame using Moors Instruments analysis software. A) Skin blood flow (flux) as a percentage of baseline (Figure 1). Black squares represent skin perfusion of mice maintained on normal anaesthetic carrier gas for the duration of the protocol.

The main concern we have with this data, and the reason we do not wish to include it in the manuscript, is an increase in baseline skin perfusion with time using an inhaled anaesthetic agent. We would like to make the reviewers aware of a number of papers highlighting the caveats of using inhaled isoflurane in conjunction with measuring cardiovascular parameters in small rodent models (C Constantinides. ILAR J, 2012; CF Yang. TC Med J 2014). However, switching the anaesthetic carrier gas to 10% oxygen (A, black circles) re-capitulated the documented peripheral vascular flush induced by hypoxia, however this was not followed by hypoxia driven peripheral vasoconstriction at later time points that we see. We feel the anaesthetic agent is masking this latter physiological response.

We also analysed K14cre- HIF2α mice with this protocol. The initial hypoxic vasodilation reflex was preserved in these mutant animals. Interestingly the degree of vasodilation was maintained throughout the remaining hypoxia protocol, indicating a significant effect of the mutant on peripheral vascular flow. We feel these data are key to the review process of this manuscript, but are compromised by the secondary effects of the inhaled anaesthetic agent, and therefore wish to include these data solely for the purposes of response to the reviewers concerns.

11) As far as we understand the authors used WT mice as controls for their conditional skin specific k.o. mice. I wonder whether the authors performed controls that demonstrate that littermates of their conditional Hif1, Hif2a and Arg1 mice show similar responses as did the WT BL/6 mice?

All procedures performed with K14cre+ animals throughout this manuscript have been accompanied by littermate control animals. The nomenclature has been clarified where appropriate. Only data for Figure 5 used wild-type C57Bl6 mice.

12) The impaired suppression of heart rate upon hypoxic exposure in K14-HIF2 k.o. is impressive (Figure 4). The authors argue that K14-Arg1 k.o. display similar findings (Figure 6). However, the figures are rather different. Therefore, we don't agree with the authors' statement that "the effect on heart rate is similar". These data rather suggest that K14-HIF2 differentially affect heart rate, which is not linked to Arg1 expression. In K14-HIF2 HIF1 might compensate for losses of HIF2. Along this line, there is some evidence that in addition to HIF2 (as shown by the authors in previous studies) HIF1 might regulate Arg1 as well (Colegio et al., Nature, 2014). Overall, we are not sure whether all the effects observed in K14-HIF2 mice are due to diminished Arg1 expression. Please clarify.

We agree that the overall response to loss of epidermal HIF-2a is broader, stronger and certainly more complex than that seen when the same tissue lacks Arg1. And we agree that this is related at least in part to the reasons cited by the reviewers. We now have revised our descriptions of these results to better reflect these considerations.

13) The authors demonstrated earlier that skin HIF-1 is required for renal EPO production (Cell 2008). Therefore, it remains unclear whether the cardiovascular responses observed in the K14-Hif1 situation are due to local regulation of tissue perfusion (as suggested in this paper) or affected by alterations in blood viscosity and blood oxygenation (due to suppressed EPO production). Experiments that dissect the contribution of EPO vs. local effects of HIF1 would help to clarify this issue (e.g. injection of rm-EPO in K14-Hif1 mice).

We appreciate the reviewers’ concerns with regard to the potential role of EPO, haematocrit and blood viscosity in the cardiovascular response observed in the K14cre-HIFα mice. The Cell manuscript referred to discusses over-expression of HIF, occurring in a K14cre-VHL deletion model; in that mutant, there is an excess of EPO and a high hematocrit. That manuscript goes on to show that basal EPO levels are unchanged in K14cre-HIFa mutant animals, and that their haematocrits are also normal.

We have measured basal haematocrits in k14cre+ mutants and littermates used here and seen no significant differences. We would also like to draw the reviewers’ attention to our recent manuscript: Cowburn, et al., PNAS 2106, where in supplemental Figure 2 we show a time course of hematocrit changes during hypoxia (10% O_2_). Although there is a small increase in hematocrit following 7 days of exposure to hypoxia, this only becomes significant following 14 days of exposure. This is consistent with the findings of many other researchers on both normobaric and hypobaric hypoxia.

14) Please better emphasize the novelty (specifically the mechanistic insights) of the present study, compared to the cardiovascular characteristics of the mutant mice reported previously by the same group (PNAS, 2013; (Cowburn et al., 2013).

We thank the reviewer for this suggestion, and in this revision have attempted to better address the novelty of this study. As regards the previous study, the key difference here is that initial study looked at static blood pressures, and did not employ telemetry, but rather described single observations of blood pressure. Additionally, and more importantly, that study did not look at response to any stressor, but looked instead at steady state blood pressures. It is the difference in a dynamic response to hypoxia that is the key novel aspect of this study, and this was not addressed at all in our previous work.

15) Cardiovascular control is a highly integrative process and peripheral resistance is one of major determinates of blood pressure. How do you integrate your finding to the overall control of blood pressure?

We agree that this is, in the end, the most important consideration: how does this finding integrate with a classical understanding of blood pressure regulation and response to stress. We feel that this will require, first, more work; and then a review article or a series of them to begin to integrate understanding of peripheral responses and their influence on the cardiovascular stress response. We hope that this article ultimately is part of the beginning of this reconsideration of how blood pressure regulation and response functions.

16) The authors need to explain whether the delayed cardiovascular responses in K14Cre-HIF-1α mice are due to higher baseline blood pressure. Likewise, changes in heart rate responses in K14Cre-HIF-2α mice might as well be due to baseline bradycardia.

We see variations in response over time here; that is, indeed, our key finding. It is these variations that demonstrate that we are not simply seeing the effects of a mild constitutive hyper- or hypotensive state on hypoxic response. These dynamic changes have considerable complexity, and will need further work to be mechanistically defined. However, we feel that they confirm our hypothesis, i.e., that there is a significant role for peripheral tissue response via HIF in the dynamic response of the cardiovascular system to changes in oxygen availability.